# Immune-Mediated Effects of Microplanar Radiotherapy with a Small Animal Irradiator

**DOI:** 10.3390/cancers14010155

**Published:** 2021-12-29

**Authors:** Soha Bazyar, Edward Timothy O’Brien, Thad Benefield, Victoria R. Roberts, Rashmi J. Kumar, Gaorav P. Gupta, Otto Zhou, Yueh Z. Lee

**Affiliations:** 1Department of Radiation Oncology, University of Maryland, Maryland, MD 21201, USA; soha.bazyar@umm.edu; 2Department of Physics and Astronomy, The University of North Carolina, Chapel Hill, NC 27514, USA; etobrien@email.unc.edu; 3Department of Radiology, The University of North Carolina, Chapel Hill, NC 27514, USA; thad_benefield@med.unc.edu; 4School of Medicine, Duke University, Durham, NC 27710, USA; victoria.roberts@duke.edu; 5Medical Scientist Training Program, The University of North Carolina at Chapel Hill, Chapel Hill, NC 27514, USA; rashmi_kumar@med.unc.edu; 6Department of Radiation Oncology, The University of North Carolina at Chapel Hill, Chapel Hill, NC 27514, USA; gaorav_gupta@med.unc.edu; 7Department of Applied Physics Sciences, The University of North Carolina at Chapel Hill, Chapel Hill, NC 27514, USA; zhou@email.unc.edu; 8Lineberger Comprehensive Cancer Center, The University of North Carolina at Chapel Hill, Chapel Hill, NC 27514, USA; 9Biomedical Research Imaging Center, The University of North Carolina at Chapel Hill, Chapel Hill, NC 27514, USA

**Keywords:** spatially fractionated radiation therapy, microplanar radiation therapy, microbeam radiation therapy, combined radio-immunotherapy

## Abstract

**Simple Summary:**

Half of all cancer patients receive radiation therapy during their course of treatment. The destructive effects of radiation on normal tissue causes side effects that significantly affect the patient’s quality of life. Microplanar radiation therapy (MRT) is a novel, promising method that, in animal models, has shown fewer effects on normal tissue and better therapeutic efficacy. MRT does not expose the entire tissue to intense radiation, but has micron-scale “valleys” between the peaks of radiation. We have previously developed an accessible method to apply MRT. Using our approach, here, we show that MRT greatly enhanced the response of the immune system to the tumor by stimulating specific signaling molecules. Moreover, we show that combining MRT with immune checkpoint therapy was even more effective in reducing the treated tumors, possibly pointing towards using similar approaches in the clinic.

**Abstract:**

Spatially fractionated radiotherapy has been shown to have effects on the immune system that differ from conventional radiotherapy (CRT). We compared several aspects of the immune response to CRT relative to a model of spatially fractionated radiotherapy (RT), termed microplanar radiotherapy (MRT). MRT delivers hundreds of grays of radiation in submillimeter beams (peak), separated by non-radiated volumes (valley). We have developed a preclinical method to apply MRT by a commercial small animal irradiator. Using a B16-F10 murine melanoma model, we first evaluated the in vitro and in vivo effect of MRT, which demonstrated significant treatment superiority relative to CRT. Interestingly, we observed insignificant treatment responses when MRT was applied to Rag^−/−^ and CD8-depleted mice. An immuno-histological analysis showed that MRT recruited cytotoxic lymphocytes (CD8), while suppressing the number of regulatory T cells (Tregs). Using RT-qPCR, we observed that, compared to CRT, MRT, up to the dose that we applied, significantly increased and did not saturate CXCL9 expression, a cytokine that plays a crucial role in the attraction of activated T cells. Finally, MRT combined with anti-CTLA-4 ablated the tumor in half of the cases, and induced prolonged systemic antitumor immunity.

## 1. Introduction

Radiation therapy, or radiotherapy (RT), is used in approximately half of all cancer patients during their treatment course [1]. Consequently, any improvements in this modality would benefit a large number of patients. The treatment efficacy of RT mainly depends on the total radiation dose given, balanced by the damage the RT causes to the surrounding healthy tissue. Most recent clinical studies have demonstrated that normal tissue damage occurs even after advanced RT approaches, namely, proton therapy [2].

Spatial fractionation techniques have shown promising results in sparing the normal tissue [3]. Microplanar radiation therapy (MRT) delivers hundreds of grays of radiation in spatially fractionated quasi-parallel micrometer planes. MRT consists of high-dose irradiation beams (peaks), separated by wider non-irradiated regions (valleys) that receive the scatter dose (Figure 1). Interestingly, preclinical studies on radioresistant orthotopic tumors have consistently found selective tumoricidal and normal tissue sparing advantages of this novel technique [4,5,6,7,8,9,10,11,12,13,14]. The vast majority of MRT studies have been conducted at four national synchrotron laboratories around the world. Limited access to these facilities is the major obstacle to the clinical translation of this method, and has also minimized the chance of reproducing the data. Most recently, we have adopted a commercial animal irradiator for in vitro and in vivo MRT studies, through the use of a high-precision multi-slit collimator. We found the treatment superiority of this approach relative to conventional radiation therapy (CRT) in a radioresistant murine melanoma model [15].

Several hypotheses have been developed to explain the wider therapeutic index of MRT, relative to CRT. Among the most interesting is the idea that the spatially fractionated pattern of MRT induces a more effective immune response against the tumor. A genome-wide comparison of MRT vs. CRT showed the differential expression of immune response regulatory genes, and several other studies have pointed to a more robust antitumor immune response subsequent to MRT [16,17,18,19]. In the present study, we show that the improved utility of the MRT approach is dependent on an intact adaptive immune system, and implicates downregulation of Treg cells, and CD8 and B-cell immune responses. Thus, MRT may, itself, help to mitigate the immunosuppressive tumor microenvironment to an immune responsive one, and, in combination with an immune checkpoint inhibitor, even greater tumor suppression was achieved.

## 2. Materials and Methods

An abstract of methods is illustrated in Figure 2.

### 2.1. Cell Culture and Clonogenic Assay

B16-F10 cell line was purchased from the Lineberger Comprehensive Cancer Center Tissue Culture Facility, at the University of North Carolina at Chapel Hill (UNC-CH). B16-F10 is a poorly immunogenic C57BL/6-derived melanoma cell line. All the studies were performed with the cells within 5 passages. Cells were cultured at 37 °C and 5% CO_2_ in Dulbecco’s modified Eagle’s medium supplemented with 100 UmL^−1^ penicillin and 100 µg·mL^−1^ streptomycin all from Corning Inc. (Corning, NY, USA) and 10% FBS (Serum Source International, Charlotte, NC, USA). In vitro dose responses studies of B16-F10 cell line were evaluated using the clonogenic assay with delayed plating after treatment as described previously [15].

### 2.2. Mice

Five- to six-week-old C57BL/6 and B^6.129S7^-Rag^1tm1Mom/J^ mice were obtained from Jackson Laboratory (Bar Harbor, ME, USA) and maintained under pathogen-free conditions. All the experiments were performed according to approved protocols by UNC-CH Institutional Animal Care and Use Committee.

### 2.3. Survival and Tumor Response Analysis and Reagent

On day 0, each mouse was injected subcutaneously on the right thigh with 8 × 10^4^ B16-F10 cells mixed with 100 μL Hanks’ balanced salt solution (Corning Inc., Corning, NY, USA). The cell maintenance, sample preparation and inoculation were performed using the protocol of Overwijk et al. [20]. One week later, mice were randomly assigned to various treatment groups as indicated (see tumor growth curve for detailed number of mice in each group, at least 5 mice/group, otherwise specified). The perpendicular tumor diameters were measured using calipers. Tumor volume was calculated using the formula L × W^2^ × 0.52, where L is the longest dimension and W is the perpendicular dimension. The overall survival was evaluated using the Kaplan–Meier method. Mice were humanly sacrificed when the tumor burden reached 1.5 cm^3^, to decrease the morbidity. For re-challenge studies, if the mice survived the first study endpoint, the same number of cells were injected into left thigh and the mice were followed-up for 60 days.

Anti-mouse CD8a (BE0004-1, BioXCell, West Lebanon, NH, USA) was injected i.p. on day 2, 0, and then twice per week for the duration of the experiment (0.2 mg per mouse per injection) [21]. For combined radio-immunotherapy, 0.2 mg of anti-CTLA-4 (9H10, BioXCell, West Lebanon, NH, USA) was injected i.p. every 2 days, starting 2 days prior to RT, for a total of 3 doses [21]. Shams received an equal volume of normal saline i.p. For all in vivo experiments, studies were concluded on day 60 post inoculation.

### 2.4. Radiation and Dosimetry

For all experiments, a commercial irradiator (X-RAD 320, PXi, North Branford, CT) was utilized. MRT was delivered with a collimator (44 identical beams; beam FWHM = 246 ± 32 μm; center-to-center = 926 ± 23 μm; peak-to-valley dose ratio at entrance = 24.35 ± 2.10; collimator relative output factor = 0.84 ± 0.04). The detailed specification of the irradiator and full dosimetric characteristics of our method are the same as reported before [15].

On day 8, the mice in CRT and MRT groups underwent radiation as previously described [15]. Briefly, anesthesia was induced by 3–4% isoflurane and maintained by 1–2% isoflurane in medical-grade oxygen at 0.8–1 L min^−1^ flow rate. Except for the radiation field, the whole animal body was shielded by 1 cm thick lead. The anesthetized mice were positioned on a dedicated mouse holder and their head, body and right hind limb were fixed. Radiation therapy was delivered to the 1.5 cm × 1.5 cm field (dose rate = 4.8 cGy.s^−1^; energy = 160 kVp; focal-to-surface distance = 37 cm). Skin isoeffective doses were measured as explained in a prior study [15]. Briefly, we used the maximum doses of CRT and MRT that did not cause any acute skin toxicity up to 30 days post radiation (MRT=150 Gy and CRT=15 Gy).

Gafchromic^®^ MD-V2 films (Ashland, Bridgewater, NJ, USA) were placed at the entrance and exit plan to confirm the dose delivery and for dosimetry measurements. The films were scanned and analyzed as before [15].

### 2.5. Immunostaining of Tumor Sections

Two days and one week after MRT, CRT or mock treatment the mice were humanely sacrificed and the tumors were harvested for histologic analysis (5 mice per group per time point). Extracted tissues were fixed in formalin for 48 h, processed, embedded in paraffin and serially sectioned into 5 µm thickness. The detailed protocol for each immunohistochemical staining can be found in Method S1. Stained slides were digitally imaged at ×20 magnification using the Aperio ScanScope XT (Aperio Technologies, Vista, CA, USA). Cells were stained for CD4, CD8a, CD45R/B220, FoxP3 and F4/80 and analyzed using the Aperio Cytoplasmic V2 algorithm. Adjustments for stain optical densities were made to ensure removal of melanin from the analysis. Default thresholds for 0, 1+, 2+, and 3+ staining intensities were used. To reduce the false-positive rate, only the cells that were scored ≥2+ were considered positive.

### 2.6. Real-Time Quantitative PCR (RT-qPCR)

Tumor cells were cultured as described in Section 2.1, seeded in 12.5 cm^2^ cell culture flasks (500,000 cell per 10 cm^2^), and left overnight for attachment. When they reached 30–40% confluency, they received 15, 25, 50 or 100 Gy of CRT or 50, 100 or 150 Gy of MRT (or sham irradiation). The cells were re-fed with new media 24 h after irradiation and all cells were harvested for RNA isolation 3 days post irradiation. Total RNA was reverse transcribed into cDNA using Thermo Scientific Maxima First Strand cDNA Synthesis Kit for RT-qPCR (Waltham, MA, USA) according to the manufacturer instructions. RT-qPCR was performed using Applied Biosystems QuantStudio™6 Flex Real-Time PCR System (Foster City, CA, USA). Murine primers and reaction conditions were obtained from the PrimerBank (Massachusetts General Hospital Primer Bank; see Appendix A for the complete list of primers). The data were normalized to the housekeeping gene b-actin and presented as fold induction over mock-irradiated cells using the delta-delta CT method. RT-qPCR analyses were performed in triplicate. Data were compared using *t*-test between CRT15 and MRT150 groups, as these are the doses used for in vivo studies; *p* < 0.05 was considered statistically significant.

### 2.7. Statistical Analysis

Statistical analyses were performed using SAS/STAT^®^ version 9.4 (SAS Institute Inc., Cary, NC, USA). *p*-values > 0.05 were considered inconclusive.

Survival time was compared between treatments using the log-rank test. An overall test of survivor equality was conducted. Significant *p*-values were adjusted using the false discovery rate (FDR) method to control the type 1 error rate [22]. Adjusted *p*-values <0.05 were considered evidence that survival differed by pairwise comparison.

To evaluate tumor growth by treatment, random coefficient models were used, modeling tumor growth as a function of time (in days). A quadratic trajectory for each individual was assumed. A compound symmetric R-side covariance term was added to the tumor growth models to account for the fact that observations for each mouse at different timepoints are correlated. Whenever the models detected the difference between treatment groups, significance regions, i.e., intervals of days for which there are significant treatment differences, were computed.

To analyze the histological samples, the *p*-values for the number of infiltrated cells in various treatment groups were calculated using the Exact Savage Multisample test, a nonparametric test appropriate to use with >2 groups. Significance was found by adjusted *p*-value, and the Dwass–Steel–Critchlow–Fligner method was employed to determine which groups were different.

## 3. Results

### 3.1. MRT, but Not CRT, Significantly Hinders Tumor Growth and Extends the Survival of Melanoma Mouse Model

In a previous study, we compared the therapeutic effect of these doses on the B16F10 cancer model, an established radioresistant murine melanoma model [21]. With a limited sample size, MRT appeared to be much more effective in reducing the tumor size, relative to CRT. The present study was undertaken to first test whether the superiority of MRT was reproducible in a much larger sample size, and then to test the involvement of the immune system in that effect.

Previously, using a clonogenic assay, we calculated the radiobiological equivalent dose of MRT vs. CRT in vitro [15]. However, the CRT survival curve was adopted from Twyman-Saint Victor, C. et al. [21]. Here, we calculated survival curves with a larger number of samples in both groups (Figure 3A), which confirms the similarity of the CRT survival curve to the adopted data.

Our previous work has shown that, in healthy adult C57BL/6 mice, the maximum dose of MRT that did not destroy the skin integrity up to 30 days post radiation was 150 Gy [15]. However, the CRT radiobiological equivalent dose (RBED) was very toxic to our mouse model, which confirms the wider therapeutic window of MRT. Based on these findings, we performed a skin assay, and used the isoeffective dose of each modality that did not induce an acute skin reaction in our mouse model, as described before [15].

We observed that CRT did not induce any significant therapeutic effect compared to mock radiation (see Figure 3B,D–F). In contrast, MRT significantly suppressed tumor growth and extended the survival of the mice inoculated with B16-F10 (*p* < 0.001). No significant differences in weight or pretreatment tumor volume were found among the treatment groups. Interestingly, one of the MRT-treated mice reached the survival endpoint without any tumor growth.

### 3.2. Intact Adaptive Immune System, CD8 Lymphocytes in Particular, Are Required for Treatment Response to MRT

To evaluate the role of the immune system in MRT, Rag^1K0^-B6 mice were employed. This model is homozygous for the Rag^1tm1Mom^ mutation, and lacks mature T and B cells. In this model, using the same dose regimen as that used with the C57BL/6 mice, we observed no significant difference in the effect of MRT or CRT as compared to mock-treated mice (Figure 3C,G–I). We also observed that the effectiveness of MRT treatment itself was also significantly decreased, relative to the immunocompetent model (Figure 3C,F vs. Figure 3I). These observations strongly implicate the adaptive immune system in the enhanced effectiveness of MRT.

To identify the most essential part of the adaptive immune system in MRT therapy, we first tested whether cytotoxic T cells (CTLs) were necessary, as their role in CRT-induced antitumor immunity is well established [23,24,25]. In contrast to immunocompetent models, but similar to the Rag^−/−^ immunosuppressed model, MRT in CD8-depleted mice neither significantly extended the survival nor suppressed the tumor growth rate compared to sham (Figure 3J–M), demonstrating the necessity of active CD8 cytotoxic cells in the enhanced effectiveness of MRT.

### 3.3. MRT Recruits Robust Antitumor Immune Cells

B16-F10 is a poorly immunogenic radioresistant carcinoma, and, thus, serves as a useful baseline for the immune system simulation of RT [20,21]. To evaluate the tumor immune microenvironment for immune surveillance, the tumors were extracted at 48 h post treatment and stained for F4/80 (macrophages). One-week post radiation treatment, the extracted tumors were stained for CD4 (helper T cells), CD8a, CD45R/B220 (B cells), and FoxP3 (Tregs). The numbers of cells in various treatment groups are demonstrated in Figure 4. The number of cytotoxic T cells and B cells were significantly higher 1 week after MRT (see Figure 4). When compared to the mock-treated mice, CRT significantly increased the number of Treg cells infiltrated into the tumor, while no difference was observed between the MRT- and mock-treated groups. Consequently, MRT resulted in a significant increase in the CD8/Treg ratio. No significant differences were observed in the number of macrophages (F4/80) and CD4 cells between the various treatment groups.

### 3.4. MRT Alone Does Not Induce Persistent Antitumor Memory

To investigate the effect of MRT on long-term antitumor immunogenicity, the same number of B16-F10 cells were injected s.q into the left thigh of the only immunocompetent mouse in the MRT group, who survived the initial challenge. We also employed five shams to verify the method. The tumor started to grow without any delay compared to the sham mice, which rejected our primary hypothesis that MRT alone induces long-lasting antitumor immunity (data not shown here).

### 3.5. MRT Upregulates CXCL9 Expression

Sufficient T cell infiltration into the tumor microenvironment is a prerequisite for overcoming tumor resistance to checkpoint blockade [26]. Several studies have highlighted the potential of radiation to convert tumors into inflamed peripheral tissues, achieved by inducing chemokines, such as INF-β, CCL5, and CXCL9, which are involved in the recruitment of effector T cells [27,28]. We observed that MRT significantly increased the expression of CXCL9, relative to the CRT dose we used in vivo (MRT 150 Gy vs. CRT 15 Gy; *p* = 0.034). In addition, interestingly, in contrast to CRT, MRT did not saturate the CCL5 and CXCL9 pathways (see Figure 5).

### 3.6. Combined MRT and Anti-CTLA4 Therapy Orchestrates Systemic Antitumor Immunity

Observing the robust effect of MRT on the acquired immune system prompted us to test the combination effect of MRT with immune checkpoint blockade. Anti-CTLA-4 was chosen since a prior study found no significant survival benefit of this agent alone on the B6-F10 model [21]. Additionally, we did not observe any significant increase in the expression of anti-PD-L1 in MRT-treated cells. Due to the long survival of the treated mice, we decided to extend the study duration to 90 days. Remarkably, the combination of MRT + aCTLA-4 was able to ablate tumor growth in 50% of the mice (Figure 6). Moreover, the treated mice survived significantly longer that the mice treated with CRT + aCTLA-4 (*p* = 0.039 CRT + C4 vs. MRT + C4).

The mice who survived 90 days post tumor inoculation were then rechallenged by injecting the same number of cells into the left thigh. Interestingly, we observed that the tumor did not grow in any of these mice for up to 60 days post inoculation. To investigate the role of CD8 cells in systemic immunity in our model, during the second rechallenge, the mice were divided into two groups after cell inoculation; one group received anti-CD8 i.p.×3 per week, and the mock group was injected with the same volume of NS. At the end of the study, the spleens of the mice in the anti-CD8 group were harvested and stained for CD8, confirming the effect of our agent (data not shown here). We observed no significant therapeutic benefit of MRT, compared to mock or CRT, when CD8s are blocked.

## 4. Discussion

MRT is a promising preclinical radiation therapy modality, which shows many advantages over the current methods of CRT. Translation of this method has been held back due to the limited number of synchrotron facilities, where MRT was first implemented and mostly studied. We demonstrated a more accessible method for applying MRT [15]. Recently, several groups have begun to employ comparable methods for physical, biological and preclinical studies, proving the high reproducibility of this approach [29,30,31,32,33,34]. Here, the evaluations of the treatment efficacy of MRT on the murine melanoma model confirmed our previous results. We observed that high-dose CRT is not effective in treating B16-F10. This is in agreement with prior studies that did not observe any superiority, over sham radiation, of CRT up to 20 Gy on B16F10 [21]. In contrast, MRT significantly suppressed the tumor growth and extended the survival of tumor-bearing mice (Figure 3).

The effect of RT on the immune response has recently been the subject of great interest, and various aspects of the immune-stimulating potential of RT have become evident [23,35,36]. Here, we found a robust antitumor immune response effect after MRT to an extent was not observed in conventional radiotherapy (CRT). An interesting finding in our study was the spectrum of therapeutic effects of MRT (Figure 3F). This individual variation in the response to RT has been observe before [37]. Although future investigations are needed, multiple hypotheses can explain this finding. From an immunological perspective, B16F10 is a highly mutable tumor model. When compared to cultured cell, tumors separated from mice found to have 35.1% novel mutations (1078 novel mutations on average). This may be due to the mutation of mismatch repair genes and may change the tumor microenvironment [38]. Interestingly, when combined with immune checkpoint blockade, this effect is still observed, which further strengthens the hypothetical role of the individual tumor microenvironment.

Combined MRT and anti-CTLA-4 can ablate half of the tumors in our model system, and induce a long-lasting systemic immune response (see Figure 6). Although other aspects of the immunomodulatory role of MRT have been investigated before, to the best of our knowledge, this is the first demonstration of the importance of the intact immune system and, particularly, of cytotoxic immune cells in the efficacy of MRT [16,17,18,19].

The adaptive immune response, and CD8 cells in particular, play crucial roles in the antitumor immunity after CRT [23]. Here, we observed that CD8s are essential for the therapeutic effect of MRT (Figure 3). Interestingly, after MRT, a significantly higher number of cytotoxic T cells infiltrated the tumor tissue (Figure 4). Our investigations using RT-qPCR revealed that MRT, in contrast to CRT, does not saturate the expression of chemokines. A more comprehensive evaluation is needed to gain a better insight into the tumor immune microenvironment after MRT.

Previous studies on the B16-F10 model system demonstrated that a single high dose of CRT enhanced the immunosuppressive microenvironment by “exhaustion” [21]. Regulatory T cells (Treg) play a key role in this process. These cells are more radioresistant than cytotoxic T cells, so their relative number increases post radiation [39,40]. Here, we observed a significant increase in the number of Treg cells after CRT (Figure 4). Furthermore, a significantly lower number of Treg cells infiltrated the MRT-treated tumor tissue (Figure 4). We hypothesize that MRT, by its unique dose distribution, provides B cells and CD8 cells with an opportunity to survive in the “cold” regions. The survived, injured cells continue to release cytokines that enable the stimuli to remain and possibly draw more CD8 cells to the area.

Here, we found that the expression of CXCL9 significantly increased in MRT150 vs. CRT15; we found significant therapeutic differences between these two modalities with varying doses. Only the following two in vivo tumor models have been studied for comprehensive transcriptomic analysis after MRT: EMT6.5, a mouse breast cancer model, and 9L rat orthotopic glioblastoma [16,18]. In a very recent review, Trappetti et al. performed a comparative gene expression analysis between these two studies [41]. Interestingly enough, CXCL9 was one of two candidates for the unique gene signature of MRT. These additional pieces of data corroborate our findings.

Another player in the enhanced immune response of MRT may be the increased number of B cells in the tumor after MRT. The number of B cells increased significantly 7 days post MRT (Figure 4). B cells have been shown to play a pivotal role in regulating the immune responses involved in cancer. B cells can suppress diverse cell subtypes, including T cells, and can facilitate the conversion of T cells to regulatory T cells, thus attenuating antitumor immune responses [42,43,44]. B cells have been reported to adapt their immunosuppressive properties within the tumor bed, and, thereby, attenuate antitumor immune responses [45,46]. Studies suggest that B cells present antigens more efficiently than other antigen-presenting cells, by selectively presenting cognate antigens (Ags) collected through surface Ig molecules, which even allows the presentation of low concentrations of Ags to T cells [47,48,49]. In the case of weakly immunogenic tumors, such as B16F10 melanoma, B cells can promote antitumor immunity by the optimization of T cell expansion [45,50].

Recent works have shown the valuable role of combined radiation and anti-CTLA-4 in the local and systemic control of cancer [51]. Ongoing efforts are focused on finding the optimal RT protocol to be combined with immunotherapy. The key feature of spatially fractionated irradiation modalities, including MRT, in radiation-induced antitumor immunity is the unique distribution of high-dose and low-dose regions, simultaneously. Johnsrud et al. have reported the synergic effect of GRID with immune checkpoint blockades [52]. The ablative effect of synchrotron-generated MRT on B16-F10 has been reported before, though it embedded FLASH effects [53]. Here, we report the first effort to combine conventional dose rate MRT with anti-CTLA-4, which resulted in ablation of the tumor in half of the mice, and activation of a prolonged antitumor immune response. As more groups start to apply MRT using commercial X-ray tubes, more studies are needed to elucidate all the therapeutic aspects of this combined therapy.

One of the major hurdles to implementing MRT as a therapeutic option is the identification of the optimal radiation dose. Due to the distinct spatial fractionation of the X-ray beam in MRT (see Figure 1), finding the actual equivalent dose of MRT versus CRT is convoluted. In synchrotron-based MRT studies, the toxicity of the normal tissue may be more dependent on the valley region parameters, due to the following: (1) ultra-high doses of X-ray destroy all cells along the beam path, and (2) the beam size is approximately 25 μm–50 μm, spaced 200 μm–400 μm center-to-center. Consequently, most of the tissue in the radiation field receives the valley dose. However, Ibahim et al. in a study on five different cell lines, using a clonogenic assay, found significantly higher cell killing in MRT than valley dose [54]. The radiation-induced bystander effect may explain this difference [55,56,57]. To make it more complicated, in the studies using the conventional radiation tube, the beams to valley FWHM ratio is larger than with synchrotron microbeams, so higher normal tissue toxicity than at the equivalent valley dose is expected. Since we investigated flank tumors, the effect of different doses of MRT and CRT on normal skin tissue in vivo was evaluated to find the skin isoeffective CRT and MRT dose that did not induce any toxic effect on normal skin tissue [15].

## 5. Conclusions

Overall, MRT is an immensely promising novel preclinical RT modality. The most encouraging feature of this novel modality is that by employing MRT, normal tissue can be potentially spared, without compromising the therapeutic benefits of RT. In conclusion, our study revealed that MRT can initiate a cascade of innate (Tregs) and adaptive (CD8 and B cells) immune responses, which mitigate the immunosuppressive tumor microenvironment to an immune responsive one. Here, for the first time, we elucidated that adaptive immune cells, and CD8 in particular, are vital for the therapeutic effect of MRT. Further investigations to test MRT in other tumor models and to optimize the doses, as well as to evaluate the underlying mechanisms of this immunostimulatory effect, are required to provide the rationale for future clinical trials. In addition, our observations demonstrate the potential of MRT to enhance the antigenicity of tumors, which will result in opportunities to study its interplay with immunotherapies.

## Figures and Tables

**Figure 1 cancers-14-00155-f001:**
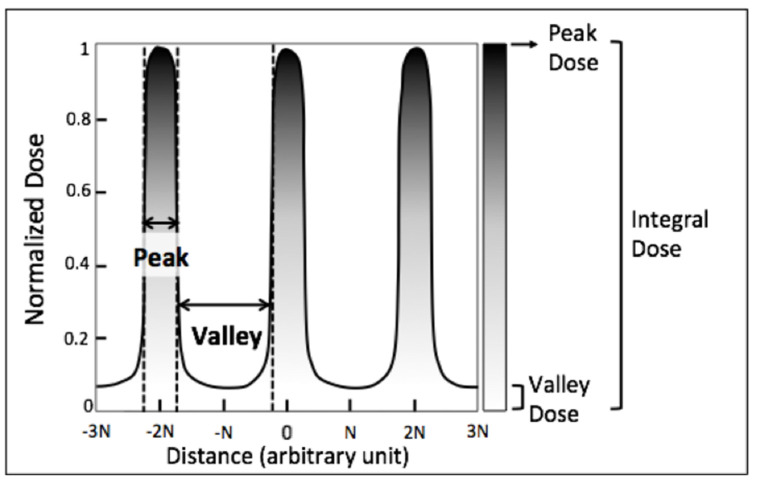
The schematic beam profile of microplanar radiotherapy (MRT). MRT delivers hundreds of Gys in spatially fractionated quasi-parallel micrometer beams. The beam profile consists of high-dose X-ray irradiation beams (peaks), separated by wider non-irradiated regions (valleys).

**Figure 2 cancers-14-00155-f002:**
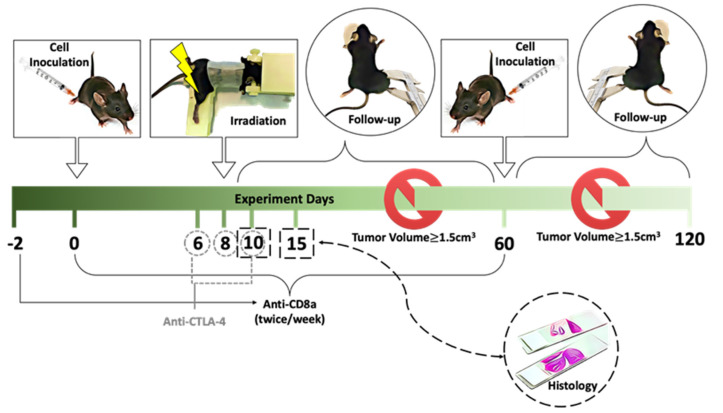
Tumor model and experiment scheme. C57BL/6 or Rag^1K0^-B6 mice were injected s.q. with B16-F10 cells (8 × 10^4^) into the right thigh on day 0. On day 8, mice randomly received radiation, either MRT, CRT or mock RT, exclusively to the tumor inoculation site with the rest of their body shielded. Anti-mouse CD8a was given i.p. on day -2, 0 and then twice a week as indicated. Anti-CTLA4 was injected i.p. every 2 days, starting 2 days prior to RT, for a total of 3 doses. Two perpendicular tumor dimensions were measured three times a week. The mice were humanely sacrificed when tumor volume reached 1.5 cm^3^. The mice survived the primary challenge, were rechallenged by injecting the same number of B16-F10 cells s.q. to the left thigh on day 60 and followed-up for the next 60 days. The study concluded on day 120. For histological evaluations, tumors were harvested 2 and 7 days after radiation.

**Figure 3 cancers-14-00155-f003:**
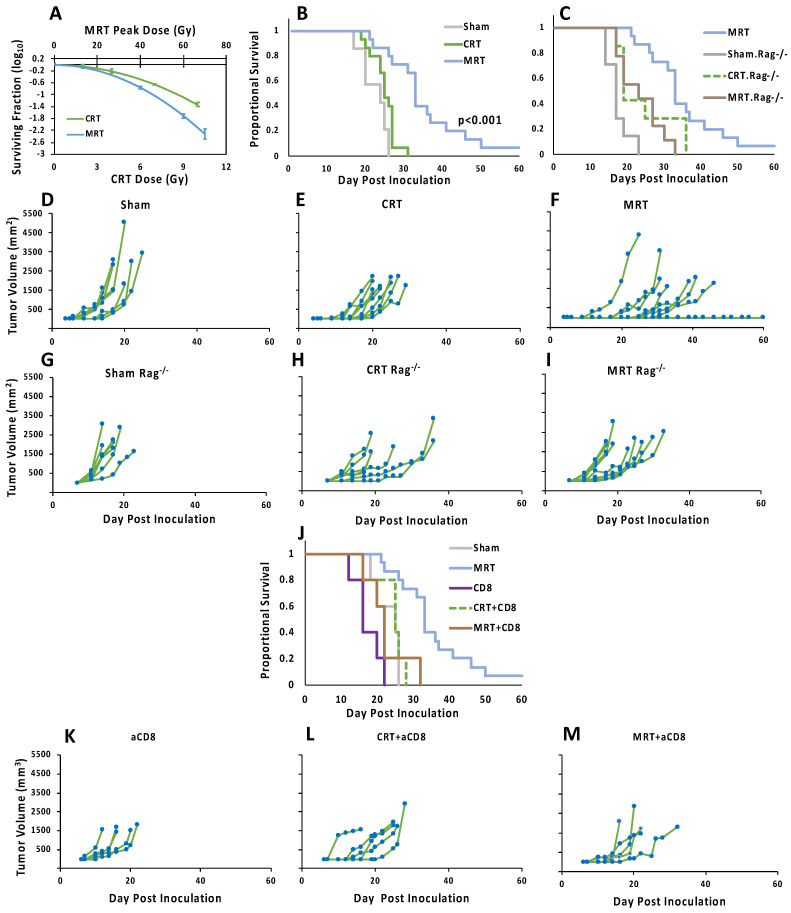
Treatment efficacy of various treatment modalities on immunocompetent and immunodeficient mouse models. (**A**) Clonogenic assay utilized to evaluate the in vitro response of B16-F10 after different doses of radiation in the following two treatment modalities: CRT and MRT. (**B**,**C**,**J**) Kaplan–Meier curves demonstrate the fraction of mice that survived at different time-points after tumor cell inoculation, the *p*-values calculated by log-rank test and adjusted using FDR. (**D**–**I**,**K**–**M**) Tumor size at different time-points after tumor inoculation in the different treatment groups, in immunocompetent, Rag^−/−^ and CD8-depleted mouse models. (**F**) One mouse in MRT group survived the primary challenge. MRT: microplanar radiotherapy; CRT: conventional radiotherapy; Rag^−/−^: model is homozygous for the Rag^1tm1Mom^ mutation, and lacks mature T and B cells; aCD8: injected with anti-CD8.

**Figure 4 cancers-14-00155-f004:**
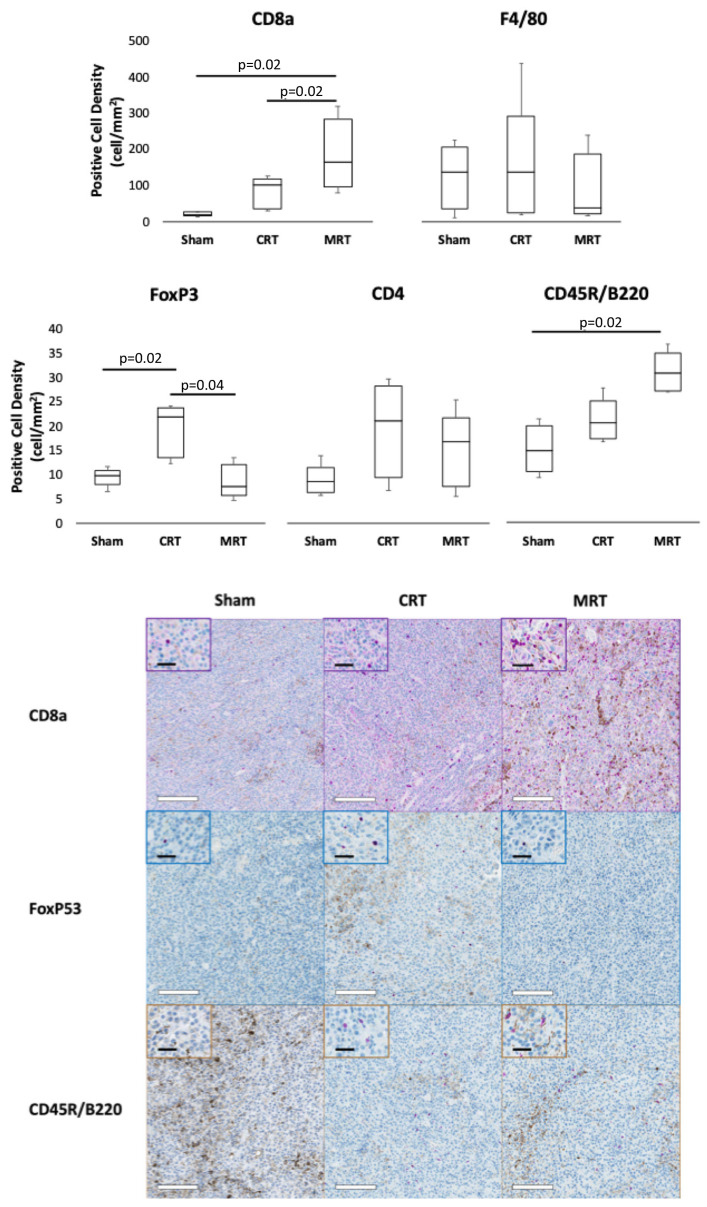
Immunohistology analysis. The box-whisker plots demonstrate the number of different immune cells in tumor tissue; *p*-values are derived using the Dwass–Steel–Critchlow–Fligner method. The tumor histology sections were stained for CD8a, FoxP53 and CD45R. White bars = 200 μm; black bars = 50 μm. MRT = microplanar radiation therapy; CRT = conventional radiation therapy; CD8a = cytotoxic T cells; F4/80 = macrophages; FoxP53 = Tregs; CD4 = helper T cells; CD54R/B220 = B cells.

**Figure 5 cancers-14-00155-f005:**
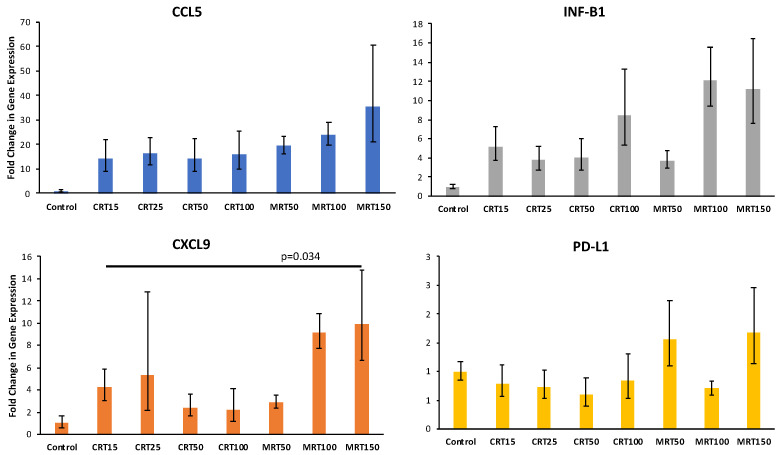
RT-qPCR analysis. The columns demonstrate the mean ± SE of fold change in gene expression in different treatment groups, evaluated using RT-qPCR. Reported *p*-value is calculated using *t*-test, comparing expression between MRT 150Gy and CRT 15Gy. MRT = microplanar radiation therapy; CRT = conventional radiation therapy.

**Figure 6 cancers-14-00155-f006:**
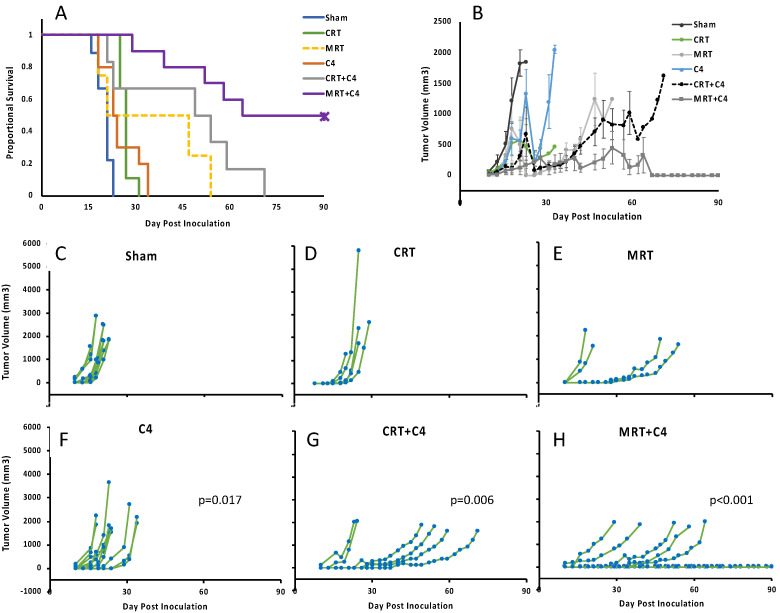
Treatment efficacy of various treatment modalities. (**A**) Kaplan–Meier curve demonstrates the fraction of the mice that survived at different time-points after tumor cell inoculation. (**B**) Median tumor volumes in different treatment group; error bars are SE. (**C**–**H**) Tumor size at different time-points after tumor inoculation in the different treatment groups, the *p*-values calculated by log-rank test and adjusted using FDR. CRT = conventional radiation therapy; MRT = microplanar radiation therapy; C4 = anti-CTLA-4.

## Data Availability

Upon acceptance for publication all the data will be available in a publicly available archive.

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
