# Peer review of "Immune-Mediated Effects of Microplanar Radiotherapy with a Small Animal Irradiator"

_cancers, 2021, doi:10.3390/cancers14010155_

Round 1

Reviewer 1 Report

The response to my main concern is not satisfactory. The authors argue that bystander effects exist and give three references with first author and year but no further information where to find them. The references are not cited in the manuscript, nor can they be found in the reference list. Even including bystander effects, it is by no means certain that they will produce a uniform cell survival across the dose profile. The authors claim that details have been added to sections 2.4, 3.1 and the Discussion section but my concerns have not been addressed in any detail and no discussion of the influence on the results has been added.

Concerning the in vivo results the authors offer variable tumour sizes as an explanation for the larger variation of the growth delay in MRT treated animals but why was this  variation in size larger in MRT than CRT treated animals ?

RBED has still not been explained.

The authors have not addressed my question regarding multiple testing and suddenly they present lower P-values without further explanation.

The authors have added a more plausible explanation for their choice of doses (15 Gy and 150 Gy), namely the absence of skin toxicity. However, this does not imply that the doses are biologically equivalent.

Author Response

-

Reviewer 2 Report

The authors have been responsive to all of the concerns and suggestions.  The paper should be a nice contribution to the growing literature on spatial fractionation and immune responsiveness.

Author Response

-

This manuscript is a resubmission of an earlier submission. The following is a list of the peer review reports and author responses from that submission.

Round 1

Reviewer 1 Report

This is a very exciting report that echoes some earlier work and recent publications regarding various types of spatial fractionation and immune response/combined effects of immunotherapy and spatially fractionated radiation.   As such it is an important addition to the literature. However, there are significant concerns about how the dose is reported,  or lack of reporting, that should be addressed.   Since the peak dose of the MRT is 10-fold higher than the chosen CRT dose, it would be very important to include what the measured or predicted valley dose is for the MRT.  This could be done using film at a minimum.  Also, what is the equivalent uniform dose for both the MRT or CRT as applied?  Is there an average dose that can be reported?    Another concern is the presentation of the data in Figure 3.  Considering the importance of Fig 3F as central to the focus of the paper, these growth plots should be larger and also should show the average tumor volume for each group.   Please add a panel that shows average tumor volume for each treatment group all on the same plot with error bars.

specific comments:

1  Overall the English is ok, but there are a number of missing descriptors and a few typos. Please review with a native speaker if possible and revise as needed.

2  The tone of the discussion could be toned down since there are other groups (Slatkin, Dilmanian, Djonov) that have looked at the immunology of microbeam therapy in various ways before.  Also, recent work with spatially fractionated photons and immunotherapy should be referenced for its similarity and general agreement with the current reported results (Johnsrud et al 2020, Radiation Research).

3   Figure 6 would also benefit by having an additional plot of only the average tumor volume over time for all groups to help show the major findings of this work.

4  In the discussion,paragraph 2, line 320, it is stated that MRT combined with checkpoint blockade ablates the tumor model.  This isn't quite supported by the figures, where it appears that nearly half of the tumors did grow back? Please consider rephrasing this.  

5  In the discussion in general, since there have been earlier studies of both immune modulating agents and MRT, consider changing the statements about 'first time' and 'first to investigate' for proper perspective on these fields.

6 some of the references, notably 31-34 are missing the author names.  please doublecheck

Author Response

Please find our response in the attached Word file. 

Reviewer 2 Report

This paper describes work looking at the benefits of MRT on immune response and inhibition of tumor growth in a mouse model system. Although the results may be potentially interesting the experimental set up, results and conclusions do not appear clear and in some cases accurate based on the number of mice used (MRT alone does not induce persistent anti-tumor memory).

Major concerns:

  • The methods and write up are difficult to follow. It is not entirely clear what was done. It looks like there were 5 mice per group injected with tumors and treated with CRT or MRT. But in 3.4 it states only 1 mouse was used, although multiple control mice were compared to it, this is not sufficient numbers to provide clear evidence of any finding.
  • There are a number of spelling issues throughout the manuscript.
  • All figures need to be improved and Fig. 4 images should be clear and large enough to see staining. In all figures X and Y axis are too small all lettering is blurry and not crisp and clear.
  • Not clear what is on Y axis for Fig. 5, should be clearly stated in the legend.
  • Figure 6 should be noted what is being compared for p values noted, and again all the lettering needs to be clearer and larger.
  • Direct repeat of section 2.5
  • Lines 157 it is not clear what is happening, where these cells are initiated from as prior to this it is only described that mice are injected with tumor cells and then tumors established in these mice are irradiated. Then all of a sudden cells are mentioned under Real-time qPCR. Where tumor cells expanded prior to RT qPCR?
  • There should be more description regarding why anti-CD8a and anti-CTLA4 were given what they expected to find and what this would tell them.
  • Figure 2 is not clear in terms of how many mice survived a primary challenge, when the primary challenge was given, etc. It should be clearly noted in the figure, currently it is not clear how there can be mice euthanized and then noted the mice that survived???
  • The technical challenges for MRT and the authors work around should be clearly noted in the discussion instead of just hinted at.
  • They conclude that more normal tissue is sparred with MRT however It is not clear in their study how they have shown that normal tissue is spared or why it would thought to have a greater chance of sparring, if one focused other radiotherapy to the tumor tissue.

Minor concerns:

  • The title does not seem appropriate as the paper does not directly interrogate a bystander effect or mechanism. Immune cell responses and tumor remission are measured but not whether if mediated by a bystander mechanism.

Author Response

(The authors gave the same response as above.)

Reviewer 3 Report

The paper deals with the immune system modification role of a new RT technology, the microplanar radiotherapy. It is a well written, interesting and important material with really new findings and conlusions. It as an experimental work, however there are important messages to the clinicans as well. 

I suggest the material for publication without any significant changes, I have only provincial recommendations.

1, There is a repetition in Chapter 2.5. and 2.6.

2, Some thoughts in the Results section could be the part of the Introduction, the Materials or mainly the Discussion (see line 192. or 197.-205. etc.) 

3, I miss a very short description of the explanation of normal tissue sparing effect of MRT from the material. This would be important to understand the alternative microenvironment immune reactios as well.  

Author Response

(The authors gave the same response as above.)

Reviewer 4 Report

This manuscript studies the role of immune system on the biological effect of microplanar radiotherapy (MRT). The use of high-dose spatially fractionated irradiation, also termed GRID, has shown promising clinical results in controlling tumour growth while sparing co-irradiated healthy tissues. The authors recently adapted an experimental small animal irradiation facility to address questions on effects and mechanisms in preclinical models. In the present work, antibodies and immune defective mice were used to demonstrate a contribution from T-cells to the growth delay in irradiated mouse tumours. Experiments were performed to support the mechanism, and an immune checkpoint inhibitor appeared to be more efficient when combined with MRT than with a 10-fold lower dose of conventional irradiation.

The experiments are appropriate to address the scientific question and, generally, the manuscript is well written. Thus the study is an important contribution to understanding this promising form of radiotherapy. However, I have concerns regarding the selection of doses based on the in vitro data. Furthermore, the explanations in the figure legends are not comprehensive.

It is argued that 150 Gy MRT is biologically equivalent to 15 Gy conventional RT. A reference is given to the previous publication (#15; Bazyar et al. 2017) which used published data from another group for comparison and new survival curves are shown in evidence. However, the authors do not consider the fact that the inhomogenous dose distribution of the MRT field will effectively inactivate all cells irradiated in the dose peaks (approx. 33%) and that the surviving fraction will be determined by the cells irradiated with much lower doses in the troughs (approx. 67%). The widths and the peak:valley dose ratios will depend on the depth below the surface  Even if the approach were considered valid, the survival curves in Figure 1A are not consistent with a 10-fold isoeffective dose ratio for the two modalities. The authors need to address this and discuss how it may influence their in vivo results.

Concerning the in vivo results, MRT seems to produce much larger variation in the tumour growth curves compared with controls or conventional irradiation so that the apparently higher efficacy relies on effects in only part of the animals whereas others hardly show any effect. What is the reason for this behavior and how does it influence the results ?

The figure legends of Figs.  4 and 5 are insufficient and need to explain the diagrams in more detail.

Furthermore, some choices of word are unconventional and should be changed. Powers of 10 should be superscript (throughout manuscript). Some minor grammatical errors (missing ‘s’ etc.) should be corrected. Section 2.6 is a copy of Section 2.5. Combined, these minor errors leave an impression of that not enough care has been put into preparing manuscript.

Detailed comments:

Line 25: add ‘e’ to read “shine”

Line 30: 'areas' should probably be 'volumes'

Line 49: it is late normal-tissue  reaction that limits dose, not tolerance.

Line 114: “industrial” is hardly the right word. Probably you mean “commercial small-animal”.

Line 171: Please give details on which values you used in the FDR method

Line 190: “inhibits” is used more commonly than “hinders”

Line 199: please explain RBED. Note that the BED concept implies a constant alpha coefficient and therefore requires that the radiation quality is the same.

Line 250: replace “demonstrated” with “shown”

Line 288: based on the curves shown in Fig. 6F,G it is difficult to believe that P=0.039. Did you correct for multiple testing ? The result relies on the very variable behaviour of individual tumours.

Line 312: A single dose of 15 Gy conventional RT is not expected to cure tumours, so this is not a surprising result.

Author Response

(The authors gave the same response as above.)
